# Automated GDPR Contract Compliance Verification Using Knowledge Graphs

Amar Tauqeer [1,2,*], Anelia Kurteva [1], Tek Raj Chhetri [1], Albin Ahmeti [1] and Anna Fensel [1,2,3]

1   Semantic Technology Institute (STI), Department of Computer Science, University of Innsbruck, 6020 Innsbruck, Austria
2   Consumption and Healthy Lifestyles Chair Group, Wageningen University & Research, 6706 KN Wageningen, The Netherlands
3   Wageningen Data Competence Center, Wageningen University & Research, 6708 PB Wageningen, The Netherlands
*   Correspondence: amar.tauqeer@sti2.at or amar.tauqeer@wur.nl

**Abstract:** In the past few years, the main research efforts regarding General Data Protection Regulation (GDPR)-compliant data sharing have been focused primarily on informed consent (one of the six GDPR lawful bases for data processing). In cases such as Business-to-Business (B2B) and Business-to-Consumer (B2C) data sharing, when consent might not be enough, many small and medium enterprises (SMEs) still depend on contracts—a GDPR basis that is often overlooked due to its complexity. The contract's lifecycle comprises many stages (e.g., drafting, negotiation, and signing) that must be executed in compliance with GDPR. Despite the active research efforts on digital contracts, contract-based GDPR compliance and challenges such as contract interoperability have not been sufficiently elaborated on yet. Since knowledge graphs and ontologies provide interoperability and support knowledge discovery, we propose and develop a knowledge graph-based tool for GDPR contract compliance verification (CCV). It binds GDPR's legal basis to data sharing contracts. In addition, we conducted a performance evaluation in terms of execution time and test cases to validate CCV's correctness in determining the overhead and applicability of the proposed tool in smart city and insurance application scenarios. The evaluation results and the correctness of the CCV tool demonstrate the tool's practicability for deployment in the real world with minimum overhead.

**Keywords:** digital contracts; data sharing; ontology; knowledge graph; GDPR compliance; smart cities; insurance

## 1. Introduction

The General Data Protection Regulation (GDPR) [1], which came into effect on 25 May 2018 across all European Union (EU) member states, lays down strict requirements for the processing, storing, and management of EU citizens' data [2,3]. The following six legal bases are defined in GDPR Art. 6 that justify the processing of personal data [3]: (i) informed consent; (ii) performance of a contract; (iii) legal obligation; (iv) protection of vital interests of the data subject; (v) performance of tasks carried out in the public interest or in the exercise of official authority vested in the controller; and (vi) legitimate interest pursued by a data controller. At the minimum, one of these six bases must be met for the lawful processing of personal data, which is defined as *"any information relating to an identified or identifiable natural person"* (Art. 4 (1), 5 and 6).

In most data sharing scenarios, organisations focus on collecting informed consent from the data subject (i.e., an identifiable natural person) (Art. 4 (1)). For instance, collecting data about an individual's online browsing behaviour is based on consent, which can be collected via cookie banners [4]. However, consent is not always enough, for example, in the case of online services where a contract is required. In scenarios such as online services

(e.g., information society services), the European Data Protection Board (EDPB) [5] also highlighted the necessity of a contract by issuing new guidelines in 2019 [6].

There are other scenarios, such as Business-to-Business (B2B) and Business-to-Consumer (B2C) data sharing, in which consent is not sufficient. This is due to the need for specific terms and conditions and their complexity. These terms and conditions yield specifications for an agreement between all contractual parties regarding what is allowed and not allowed. In addition, some of the main differences between consent and contracts are the following: (i) consent can be revoked at any time, while a contract cannot be terminated before its minimum duration; (ii) consent has predefined clauses, while contract clauses can be negotiated until an agreement is reached by all involved parties.

There is an opportunity for organisations to manage personal data under the GDPR in digital contracting (i.e., a process that transforms the entire contract lifecycle into digitalised and collaborative workflow). However, many organisations in insurance, mobility, and smart city domains still face challenges in binding GDPR rights with businesses and fail to comply with GDPR, specifically when contracts are used [7]. Another persistent challenge for many small and medium enterprises (SMEs), especially in the mobility and insurance sectors within the smart cities' domain, is the unprecedented amount of data. These data are generated every day and spread across different silos (organisations, departments, people, and databases) [8]. In our scenarios, data include contractual information, which can number in thousands per month in an organisation such as LexisNexis Risk Solutions, a leading insurance data provider. Locating specific data and permissions for its sharing, such as consent and contracts, can be time-consuming and computationally expensive due to the lack of interoperability. We can define the primary challenges regarding contracts and their management as (i) building interoperable GDPR-based contract models and (ii) performing GDPR contract compliance verification (CCV). *In digital contracting or agreements, the CCV is a process that ensures data processing according to GDPR by detecting contractual conflicts or breaches. A contract breach or conflict is a failure without legal excuses to perform any promise that forms all or part of the contract* [9]; (iii) monitoring contract execution; and (iv) updating contracts and the involved contractual parties accordingly.

While challenges vary in complexity, all should be solved in a scalable and secure GDPR-compliant manner. To do so, organisations have to adopt security measures and data protection on contracting services and processes regarding GDPR (Art. 25, 32). According to Art. 28 (3), contracts must include the following details: (i) the subject matter and duration of the processing; (ii) the nature and purpose of the processing; (iii) the type of personal data and categories of the data subject; and (iv) controller's obligations and rights. It also sets the minimum required contractual clauses, such as the duty of confidence, security measures, data subject rights, audits, and inspections. Data subject rights protection is another crucial challenge for organisations according to GDPR (Art. 25). Organisations must implement GDPR technical organisational measures (TOMs). GDPR TOMs comprise all provisions put into place to guarantee the security of personal data, such as pseudonymisation. These provisions implement data-protection principles such as data minimisation (see details in Section 4.2.1).

This paper focuses on two scenarios from the smashHit [10] project, which aims to develop a scalable, trusted, and secure solution for data sharing and contract management in the connected car and smart city domains. Use case 1 (UC1) focuses on data sharing in the insurance domain, where data sharing is key to informed decision-making. Although informed consent is the main legal basis for data sharing between the data subject and the data processor (i.e., an insurance company), when data are sold or analysed by third-party entities, additional terms and conditions must be presented and agreed upon. These form the basis of a contract, which becomes the main legal basis. Use case 2 (UC2), on the other hand, presents a data sharing scenario in the smart city domain, where an unprecedented amount of data (contractual information) is simultaneously emitted, shared, and analysed by multiple agents (i.e., software, humans, and organisations). In cases such as B2B data sharing in UC2, a contract is also needed as it provides in-depth specifications

of each contract party's obligations and the specific terms and conditions that need to be followed.

Research on GDPR compliance for digital contracts has started to gain popularity, especially in the Semantic Web domain. Solutions such as smart contracts (based on Blockchain [11,12]) and semantic contract or agreement [13–16] based on knowledge graphs (KGs) are commonly used for digital contracting. However, GDPR rights, such as the exercise of rights to the eraser and right to rectification, identifying the data controller or data processor, and data transfer are still persistent issues in smart contracts [17,18]. A further challenge with smart contracts is the classification of the various contractual parties (e.g., joint controllers) involved. The possible misclassification can directly affect the contractual party's responsibilities under the law and their potential liability for violations. These contracts define rules and penalties for an agreement and automatically enforce those contractual clauses. In digital contracting, machines do not always understand contractual terms. In such a case, smart contracts cannot handle these contractual terms that are vague [19]. GDPR compliance using Blockchain technology for data processing results in compliance issues due to the different methods used to ensure privacy-by-design and privacy-by-default [20].

KGs and ontologies can aid the building of common solutions, foster interoperability, support knowledge discovery, and decision making [14–16,21,22]. In UC1 and UC2 as discussed above, the usage of web technologies in combination with semantic technologies ensures information reusability, reliability, and inference to support the end users on the web [23]. The use of KGs for consent-based GDPR compliance has already proven to be beneficial in our previous work presented in [24]. In our earlier work [24], we proposed and developed a scalable data protection by design tools for automated compliance verification and auditability based on informed consent using KGs. This research focuses on performing GDPR contract compliance verification, where consent is not enough, for example, in online services [6]. Furthermore, in comparison to the diverse consent ontologies that are available, as shown in [8], there are few ontologies that model contracts based on GDPR. Following this and our previous work in [24], we present a KG-based solution for digital contracting, which has the following functionalities: (i) binding GDPR with data sharing contracts and (ii) performing CCV checks on contracts.

The main contributions of our work are as follows:

1. A scalable tool for managing semantic-based contracts within smart city and insurance use cases;
2. Our tool implements a KG-based approach for GDPR-compliant CCV;
3. An ontology and KG for contracts that can be reused in various cases and domains.

We would like to emphasise that our tool reduces contractual execution time and cost compared to manual contract compliance verification. With the example of the contract repository, contractors can easily track their data usage and obtain compliance notifications within the tool. Last but not least, our tool improves contract management processes, which ultimately reduces the overall contracting cost compared to the (classical) manual contracting approach or to ad hoc solutions that each come with their vendor lock-in solutions.

The rest of the paper is structured as follows. Related research studies are presented in Section 2. We describe the approach for building this tool in Section 3. The tool's architectural design and implementation are presented in Section 4. We discuss the evaluation and results in Section 5. Finally, the conclusion and future research are presented in Section 6.

## 2. Related Work

This section presents an overview of related work on contract management (Section 2.1), semantic contract modelling (Section 2.2), and on contract-focused GDPR compliance verification (Section 2.3).

*2.1. Contract Management*

Longo et al. [25] present a model for the construction and management of Service Level Agreements (SLAs) by extending the XML-based WSLA (Service Level Agreements for web services) framework [26]. Two of the main challenges that were solved include (i) the lack of standard models representing service contracts and their SLAs in service-oriented architecture (SOA) and service network environments and (ii) making SLAs effectively machine-readable.

The first one is solved by complementing WSLA with composition topologies and rules. They achieve the second one by modelling the template as a digraph that is implemented in a NoSQL (Not only Structured Query Language) [27] graph DBMS (Database Management Systems). The evaluation of the functionalities was performed based on the following five metrics; (i) availability; (ii) response time; (iii) mean time to repair; (iv) mean time to failure; and (v) mean time between failure. Based on these assessment capabilities, they offer a tool named DAMASCO (Data Manager for Service Composition) to Information Technology (IT) professionals during the design phase. However, this tool does not comply with GDPR for data processing.

Guo et al. [28] present an electronic contract management system based on Blockchain technology for commodity procurement in the electric power industry. The proposed BEcontractor process-oriented contract management system solves a series of security issues (e.g., signatures and digital certificates) existing in traditional contract management systems. The evaluation of the system has shown that it can significantly reduce the time and cost of completion of the contract signing process. With this, they also present that the payment period is shortened from three months to around one month. BEcontractor works under China's legal protection of electronic contracts, but there is no information about data processing rights, such as GDPR, and there is no information about B2B contracts.

Voronova [29] proposes a contract management system, which provides a classification of contracts (e.g., sales contract and supply contract) and their types (e.g., unilateral agreement and bilateral treaty) for network trading companies. The contract types are based on several features, such as the rights and obligations of interested parties to the contract. The author emphasises determining the contract strategy by choosing the contractual structure (based on types, sequence of conclusion, and relationship of contracts), setting the key performance indicator (KPI) of the contract, the KPI of business processes, and establishing relationships between them. The author provides guidelines for organisations to improve their efficiency and competitiveness and to protect their interests by improving the efficiency of the contract's management. However, there is no information about data processing under GDPR.

Schmidt et al. [30] propose an electronic Contract Management System (eCMS) in the health domain. The primary objective of eCMS focuses on Continuous Process Improvement (CPI) in eCMS to align best with Lean Six Sigma and Quality Management frameworks. The authors standardised the processes and increased both the system's productivity through workflow design and its efficiency and achieved improved quality with respect to the eCMS process. The Cobblestone for eCMS web-based system, which provides contract tracking, drafting, and administration functionalities, was selected. It also offers contract lifecycle management that streamlines and automates the entire contract process from contract drafting to completion. However, there is no information about how personal data are treated in the eCMS and what types of contracts are available.

Simić et al. [31] explored the applications of smart contracts in the legal domain and proposed a Blockchain-based smart contract management system with a user interface for end-user accessibility. The authors conclude that without any intermediary involvement, smart contracts can be concluded more efficiently and can reduce the contract's cost [31]. From the underlying mechanisms of the Blockchain, there are many advantages, such as no risk of data loss and malicious data manipulation arising. For potential disputes, smart contracts should provide a mechanism to resolve them fairly [31]. For these potential disputes, the contractual parties have to rely on the legal system. Despite the system being

open-access, GDPR's legal basis (necessary for lawful data processing in contracts) has not been discussed.

### 2.2. Semantic Modelling

Zou et al. [32] present a formal service contract model for cloud service and accountable Software as a Service (SaaS) by utilising semantic technologies. The model allows service providers and consumers to monitor the execution of service contracts and to keep track of obligation fulfilment during service delivery. They propose a graphical model based on Colored Petri-Nets (CPN) to model contract obligations and their interdependencies. However, this service contract implementation supports only B2C contracts and does not comply with GDPR because it was developed and implemented before GDPR enforcement.

Perrin and Godart [33] propose a semantic-based contract model to describe business interactions, deploying cross-organisational activities (called synchronisation points) and enforcing and controlling policies. A rule-based approach is used for this model. The resulting contract model describes the processing of web services for cooperation and the enforcement of contract clauses by synchronisation points. Similarly to [32], the work in [33] focuses only on B2B contracts and was conducted before the acceptance of the legislation; thus, its compliance is questionable.

Kabilan and Johannesson [34] present the Multi-Tier Contract Ontology (MTCO), which consists of three layers. The first layer defines conceptual models of contracts, while the second layer is responsible for defining specific types of contracts. The third layer defines contractual obligation and their fulfilment patterns. Furthermore, MTCO models have different stages with respect to the contract-signing process (e.g., conception, drafting, and signing), which can be beneficial for modelling contracts in detail (e.g., to provide provenance information). In addition, MTCO models contract details such as performance obligations, rules, rights, and payments. However, the ontology does not clearly differentiate between traditional contracts and electronic contracts.

Cesare and Geerts [35] present an ontology for contracts, which consists of the following three building blocks: (i) agreements amongst persons, (ii) promises, and (iii) considerations. The ontology in [35] modelling contracts includes types (e.g., verbal and written), events related to the execution, fulfilment, and the exchange of contracts. However, modelling specific contract domains (e.g., in sales) and the formalisation of the ontology in Web Ontology Language (OWL) are left as future research directions. Further, this ontology is developed before the acceptance of GDPR, and specific legislation requirements regarding data processing have not been considered.

Petova et al. [36] propose and develop Financial Industry Business Ontology (FIBO) for contracts. It is a collection of eleven separate ontologies that define entities and processes in business and finance domains. FIBO does not focus on specific laws (e.g., GDPR). However, it provides a detailed semantic model of concepts such as contracts and agreements, which can be used as a foundation for any ontology focused on GDPR. Although FIBO does not focus explicitly on GDPR when modelling contracts, recent updates regarding its mapping to the legislation have been made. Furthermore, FIBO can be helpful for the formation of new ontologies that expect to depict business and monetary ideas and can be utilised in combination with the Data Sharing Agreement Privacy Ontology (DSAP) [2] to assist information and interaction straightforwardness. We reused FIBO for many classes (e.g., *fibo-fnd-agr-ctr:MutualContractualAgreement* and *fibo-fnd-agr-ctr:Contract*) and properties (e.g., *fibo-fnd-agr-ctr:hasEffectiveDate* and *fibo-der-drc-ma:hasBeneficiary*) related to contracts.

### 2.3. Compliance Verification

Gangl [37] analyses the impact of GDPR on third-party contracts. The author conducted a survey, which can be used for an in-depth analysis of contracting parties in the domain of cloud service providers. The survey's result is compared with the purpose of the GDPR to find out whether it supports the bilateral relationship in new and disruptive

technologies. Further, they assess whether Blockchain technology might be a valid alternative to achieve GDPR compliance. They argue that Blockchain technology might be a valid alternative, but it has limitations.

Doe [38] describes guidelines for GDPR compliance verification from the perspective of the law firm sectors. The author provides a comprehensive introduction to the regulations and practicalities for law organisations in compliance with GDPR. The author makes a set of guidelines regarding the record of data processing, training needs, security, and contract documentation. There were only sets of guidelines for GDPR compliance verification, but there was no information about its implementation.

Ferrari [39] discusses data protection issues in Blockchain technology. The author has examined different aspects of Blockchain technology, which resonated or conflicted with the GDPR. For instance, GDPR is tailored to the model of centralised data storage. However, data stored on Blockchains do not fall outside its application. The author emphasises GDPR requirements, which require more tension with the structure of Blockchain technology (e.g., the right to the eraser, data minimisation, and conditions for transmission of data to third countries).

Starno et al. [40] present the implementation of a prototype for a contract compliance checker limited to B2B interactions. They describe the design and implementation of an independent third-party contract monitoring service (Contract Compliance Checker (CCC)), which provides the contract specification in force, and it is capable of observing and logging B2B interaction events while determining the business partner's consistency with contracts. They developed a contract specification language EROP (for Events, Rights, Obligations, and Prohibitions) for the CCC. This model only deals with B2B and does not comply with GDPR.

Aziza et al. [41] present a contract compliance model for Islamic Finance Knowledge (IFK) using semantic web technology. Further, they describe contract compliance rule modelling to set Islamic Finance Contract (IFC) Heuristics that can be associated with a transaction model. Three comparative studies were conducted on the competing rules of formalism. This model does not focus on specific laws (e.g., GDPR).

Pantlin et al. [42] describe the attention on emerging market practices in supplier contracts in light of GDPR compliance. The authors discuss the complexity in the supply chain for businesses due to increased outsourcing to the cloud or the third-party external service providers. Further, challenges related to supplier contracts, such as rights audits, security measures, and sub-processors, are discussed as well. Therein, we do not have discussions and guidelines for GDPR compliance on B2B contracts.

Masoud and Omer [43] present a GDPR compliance tool supporting cloud providers in the cloud-based service delivery. They introduce the encoding scheme for GDPR rules by creating legal questions, which is sotred in the Blockchain for auditing purposes. To investigate the execution cost of GDPR compliance checking, they deploy it on smart contracts in a Blockchain test network. The presented GDPR compliance tool does not comply with B2B contracts and contracts without consent.

Maria et al. [44] presented an approach for the contract compliance evaluation regarding imperfect timing information to detect violation likelihood. They describe the importance of time constraints (e.g., a time window) for performing compliance on contracts. Based on these, they construct a time contract language. They only describe the model mathematically and do not provide any implementation details with any use case or tool.

To summarise, our work builds on the related work in the field and presents an exploration into: (i) the construction and management of contracts; (ii) how KGs and ontologies can aid the building of common solutions and support knowledge discovery to ensure information reusability, reliability, and inference; and (iii) how the CCV with GDPR performs. The related work, overviewed in this section, is focused on exploring how organisations bind GDPR rights in contracts by providing guidelines [38], modelling the contracts [32–36], developing contract compliance tools [40,43,44], discussing data

protection issues in Blockchain technology [37,39], and describing contract management tools [25,28–31]. We followed the approach described in [34] to build our contract model by reusing FIBO [36], which models standard contract-related classes and properties. For implementing the CCV, we followed the CCC approach in [40] and guidelines presented in [38].

## 3. Approach

This section details the approach of our study. However, before discussing our approach, we first provide an overview of the contract's lifecycle, as this provides the bigger picture of our work and its complexity. Following the overview of the contract lifecycle, in Section 3.1, we provide details on the semantic model that is used in contract KGs. In Section 3.2, we provide details about our approach to CCV, and finally, in Section 3.3, we provide example scenarios where CCV can be used. The CCV example scenarios are based on use cases UC1 and UC2 of the smashHit project, of which this work is part of.

Figure 1 presents the contract's lifecycle management, which describes the relationship between the CCV and contract management. The contract's lifecycle consists of six stages, as shown in Figure 1.

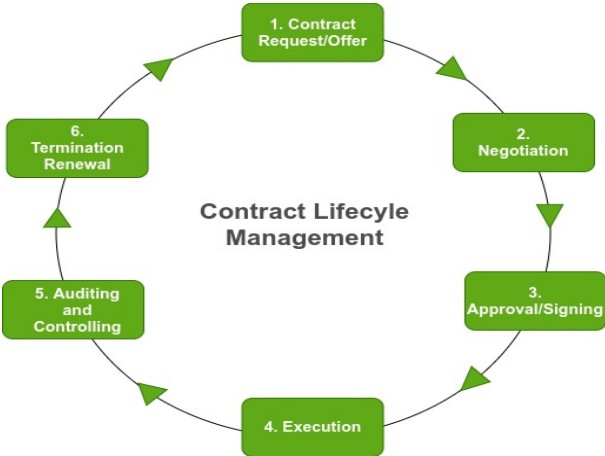

**Figure 1.** Contract lifecycle management.

A contract request or offer is the initial stage of contract lifecycle management. In this stage, the initial contract draft is created in collaboration with different departments depending on the organisation. However, it is important to understand that most contracts are not agreed upon and signed as-is. There may be substantial changes that need to be made before all involved contractors can reach an agreement. In a B2C contract, a consumer must agree and sign the contract as-is to obtain benefits from online services, for example.

After the creation of the initial contract draft, the next stage is the negotiation stage. Here, the initial draft is available for all involved contractors to review. Often, this stage is the longest and most challenging stage in a contract's lifecycle management. Contractual parties' roles (e.g., the data subject, data controller, and data processor) are defined in this stage as well. A contract contains many contractual terms and clauses, which can be defined during the negotiation stage. Depending on the number of parties involved, it can take quite a bit of back-and-forth before a final agreement can be reached.

The next stage (i.e., signing) is responsible for the signing of the contracts. In this stage, contractors have to sign the contract once they agree on all contractual clauses defined in the previous stage. The contract management software often includes useful features that allow users to route the official version of the contract to contractors and allow individuals as needed during the contract signature process. Another important aspect of contract lifecycle management is the storage and execution of contracts. The execution of the contract begins once a contract is signed. It ensures that the contracts are properly filed, organised, and able to be found easily when needed. The complexity of this stage increases

while determining the contract's storage and contract execution. All contractual states (see details in Section 3.2) become valid at this stage.

After executing the contract, the next stage is auditing and controlling. It handles the contract audits and controls the execution of CCV checks and the contract's validity. Organisations and agencies need to focus on refining this stage and ensuring compliance so that nothing slips through the cracks. Furthermore, the termination or renewal stage handles the contract's status such that either the contract will be terminated or renewed. The CCV process is mainly focused on the auditing/controlling and termination/renewal stages of the contract's lifecycle management.

### 3.1. Semantic-Based Contract Model

In order to perform compliance verification checks, the CCV tool requires contractual information, such as contractual terms, contractors, and obligations. This information comes from data sources, such as KGs. This section presents how ontologies can be used as data models in KGs and how the semantic models of contracts are constructed. The smashHitCore ontology [45] is developed to perform GDPR compliance verification checks based on consent and contracts. In this paper, we only describe and present the semantic-based contract model, while the semantic representation of consent is presented in [24].

The class *fibo-fnd-agr-ctr:Contract* (Figure 2) is used to model contracts and contractual obligations. In the context of use cases UC1 and UC2, a contract should present all of the necessary information for one to make an informed decision. However, the semantic model itself should also be generic enough so that it can be reused for various contracting scenarios.

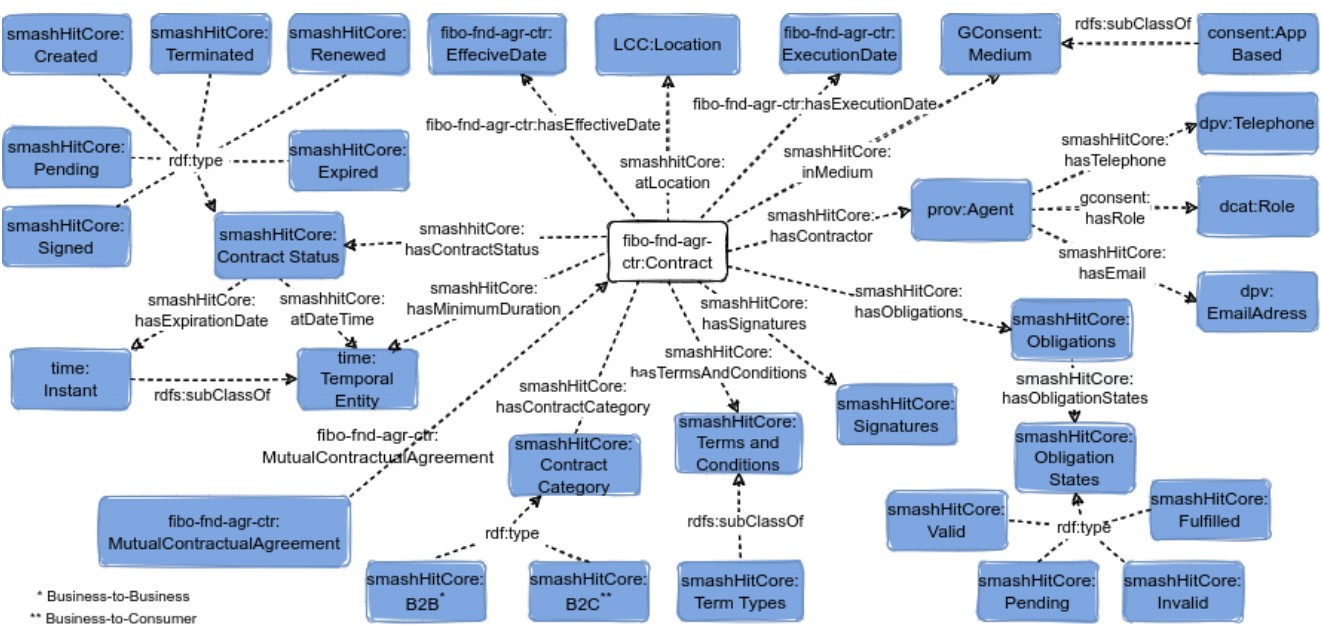

**Figure 2.** Semantic representation of contracts in smashHitCore.

To cover both UC1 and UC2, we have reused *fibo-fnd-agr-ctr:MutualContractualAgreement* from FIBO [36], which is generic enough to cover both use cases. A mutual contractual agreement involves an exchange of promises in which the promises made by each party represent considerations supporting the promises of the other party. Two categories of contracts have been modelled—*smashHitCore:BusinessToBusiness* and *smashHitCore:BusinessToConsumer*—according to UC1 and UC2.

A contract can be associated with a specific contractor via the object property *smashHitCore:hasContractors*. Specific terms and conditions can be related to a contract via the

*smashHitCore:hasTerms* and *smashHitCore:hasObligations* object properties of class *fibo-fnd-agr-ctr:Contract*. Once a contract is signed (*smashHitCore:Signed*), the obligations associated with it become active (i.e., all contractors need to start adhering to them). If a contract has expired, then all obligations become invalid. To capture this information, we have modelled different obligation states with the class *smashHitCore:ObligationState*, namely *smashHitCore:Invalid*; *smashHitCore:Valid*; *smashHitCore:Pending*; and *smashHitCore:Fulfilled*. A contract has different object properties such as *fibo-fnd-agr-ctr:hasContractualElement* (e.g., terms and conditions) and *fibo-der-drc-ma:hasBeneficiary*. To differentiate between the date when a contract is created (i.e., all agents agree upon a set of terms and conditions and a policy) and the date when a contract becomes effective, we reused properties *smashHit-Core:hasCreationDate* and *fibo-fnd-agr-ctr:hasEffectiveDate* accordingly. The property *smashHit-Core:hasExpirationDate* refers to the date a contract expires, while *smashHitCore:hasEndDate* can be used in cases when a contract is terminated before its expiry date. To ensure the integrity of contracts, we defined the object property *smashHitCore:hasSignature*, which can be used to store the signatures of all contractors of a specific contract. Information about the used prefixes is available in Appendix A.

After presenting the semantic model of contract, we now describe the CCV in Section 3.2.

### 3.2. CCV

Figure 3 shows a general overview of the CCV process, where a data source (e.g., KGs) is used as input. The first step in the CCV process is to extract contractual information from the data source. Data sources can vary and depend on the organisation. In the current scenario, we use KGs as a data source to store contractual information. We check the category of contracts in the second phase of the CCV process. The approach supports not only B2B and B2C contracts but also consent-based contracts. In the third phase, consent-based validation performs on each contract. In CCV, a validation check is performed to validate the contractual clause in the fourth phase to obtain violation or expiration results. These results are presented in the fifth phase of the CCV process. The contract status and clause state are updated in the KGs with violation or expiration results. In the last phase of the CCV process, contract violation or expiration notifications are sent to contractual parties.

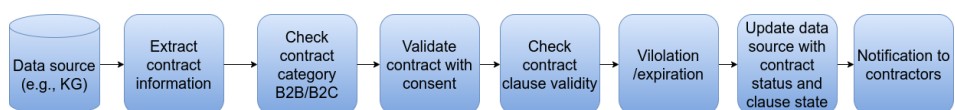

**Figure 3.** A general overview of CCV process.

After discussing the general overview of the CCV, we now describe the relationship of the contract breaches with GDPR based on UC1 and UC2. To illustrate, let us assume we have two organisations: LexisNexis and Infotripla Oy. The first acts as a data controller, whereas the second acts as a data processor according to GDPR. In data processing, where a contract is required, it must satisfy the requirements defined by GDPR (e.g., Art. 28, 32). For instance, the data processor must notify the data controller if there is a breach of a contract. The insurers can view the information about the data storage and its usage. Personal information needs to be anonymised by the tool. The tool must also satisfy the TOMs defined by the GDPR (see detail in Section 4.2.1). Complying with GDPR compliance, our tool fulfils all these requirements. It performs compliance verification checks, ensuring a contract breach, control over all running contracts, updating the contractual parties about the contract statuses, and GDPR compliance verification.

The contract dates (i.e., start date, effective, and end date), status, and clause states are key factors in performing contract validity checks. These contract dates comprise creation, effective or execution, and end date. The value of contract status depends on changes in contract dates and clause states values. These clause states are associated with contractual clauses. In the negotiating process, these states become active once a contract is signed.

We explore contractual clauses and the clause states to illustrate contract breaches. As an example, we can write a contractual clause as a tuple, as shown as follows:

$$clause(s, a, o, [ts, te])$$

where s = subject; a = action; o = object; ts = start time; and te = end time.

The contractual clause states comprise *Invalid*, *Fulfilled*, *Pending*, and *Violated* [46]. A contractual clause becomes invalid if the end time is already passed when it is assigned. The contractual clause is said to be fulfilled if it has been assigned and its action has been carried out its activities during the time window [ts, te]. If a contractual clause has been assigned, has not been fulfilled, and is not invalid but has an end time that is passed, then it is violated. If a contractual clause is not invalid and has not yet become fulfilled or violated, then it is pending. We explore such a clause with the following example. Suppose we have the following.

$$cl_1 = obl(Bob, submitreview(Bob, p1), [16/02/22, 25/02/22])$$

$$cl_2 = obl(Bob, submitreview(Bob, p2), [16/02/22, 25/02/22])$$

There are two contractual clauses in tuples $cl_1$ and $cl_2$, where Bob has to submit two reviews for the papers p1 and p2 within a specified time framework. If Bob submits his review for p1 on 25 February, $cl_1$ becomes fulfilled. If the tuple $cl_2$ has started, its status becomes pending until 25 February 2022. Tuple $cl_2$'s status becomes violated if Bob does not submit a review for p2 on 25 February 2022. A contract has many contractual terms that define contractual clauses. Contract status changes due to these contractual clause states. A contractual clause with violations also changes the contract's (associated with that clause) status to violate. The tool sends a notification about these violations to the contractual parties. We formalise the logic we discussed here in order to make the presentation of rules more clear and more concise. For the sake of simplicity, we omitted existential quantifiers for unbound variables. Hence, we have obtained the following rules: (i) The first rule states that a clause will have a pending state if it does not have any of the other clause states, such as fulfilled, invalid, or violated; (ii) the second rule states that if a clause is pending and has an associated obligation that is set in the past, then the clause is automatically set to invalid; (iii) the third rule states that if a clause with a pending state that has an associated obligation which has not been submitted until the end date, then it is violated; (iv) the fourth rule states the opposite compared to the previous rule, in the sense that a pending state becomes fulfilled if all associated obligations are submitted within the required period; finally, (v) the last rule states that if a clause is violated, then the corresponding contract is also violated.

$$\forall X \forall Y \ Contract(X) \ \wedge \ hasClause(X, Y) \ \wedge \ Clause(Y)$$
$$\wedge \ \neg hasState(Y, fulfilled) \ \wedge \ \neg hasState(Y, invalid) \ \wedge \ \neg hasState(Y, violated)$$
$$\rightarrow hasState(Y, pending)$$
$$\forall X \forall Y \ Contract(X) \ \wedge \ hasClause(X, Y) \ \wedge \ Clause(Y) \ \wedge \ hasState(Y, pending)$$
$$\wedge \ hasObligation(Y, S, A, O, time_s, time_e) \ \wedge \ time_{assign} > time_e \rightarrow hasState(Y, invalid)$$
$$\forall X \forall Y \ Contract(X) \ \wedge \ hasClause(X, Y) \ \wedge \ Clause(Y) \ \wedge \ hasState(Y, pending)$$
$$\wedge \ hasObligation(Y, S, A, O, time_s, time_e) \ \wedge \ time_{curr} > time_e \rightarrow hasState(Y, violated)$$
$$\forall X \forall Y \forall S \forall A \forall O \forall time_s \forall time_e \ Contract(X) \ \wedge \ hasClause(X, Y) \ \wedge \ Clause(Y)$$
$$\wedge \ hasState(Y, pending) \wedge hasObligation(Y, S, A, O, time_s, time_e) \ \wedge \ time_{curr} <= time_e$$
$$\rightarrow hasState(Y, fulfilled)$$
$$\forall X \ Contract(X) \ \wedge \ hasClause(X, Y) \ \wedge \ Clause(Y) \ \wedge \ hasState(Y, violated)$$
$$\rightarrow hasState(X, violated).$$

Let us consider a running contract $c_1$ with the associated clause $cl_1$. After the application of the first rule, we obtain the following.

$$Contract(c_1) \wedge hasClause(c_1, cl_1) \wedge Clause(cl_1) \rightarrow hasState(cl_1, pending)$$

Given $time_{curr} = 27/2/2022$, and after the application of the third rule, we obtain the following.

$$Contract(c_1) \wedge hasClause(c_1, cl_1) \wedge Clause(cl_1) \wedge hasState(cl_1, pending)$$
$$\wedge \ hasObligation(cl_1, bob, submitreview(bob, p_1), p_1, 16/2/22, 25/2/22)$$
$$\wedge \ 27/2/22 > 25/2/22 \rightarrow hasState(cl_1, violated)$$

Finally, given the last rule from above, for the contract, we also deduce the following.

$$Contract(c_1) \wedge hasClause(c_1, cl_1) \wedge Clause(cl_1) \wedge hasState(cl_1, violated)$$
$$\rightarrow hasState(c_1, violated) \quad \square$$

After an illustration of a contract breach with examples and rules, we describe CCV process scenarios in Section 3.3.

### 3.3. CCV Scenarios

This section presents the CCV overview with four scenarios based on UC1, UC2, and industrial requirements discussed in Section 1. Before discussing the scenarios, we present the overview of the CCV process (see Figure 4), which shows a distinction between each scenario with a specific colour. Second, we describe each scenario in more detail in Sections 3.3.1–3.3.3.

Figure 4 presents the overall CCV process comprising scenarios, such as B2C or B2B contracts, consent-based B2C contracts, and consent-based compliance verification on B2B contracts. It presents the extraction of all contractual clauses from the KGs via the SPARQL (Simple Protocol and RDF (Resource Description Framework) Query Language) [47] endpoints. A contract repository may have multiple contractual clauses that need multiple iterations for validation. Each contractual clause has a contractual party, *contractID*, *stateID*, *termID*, and a time window (including start and end time). The contractual clause state and contract status require validating each contractual clause. In CCV, only contracts having created or signed statuses and contractual clauses with a pending state are involved. The information about a contract's status and contractual states is extracted from the KGs. With this information and the current date, we can validate contractual clauses.

Figure 4 is divided into two blocks, namely block-1 and block-2. Block-1 shows the following three scenarios used to perform CCV: (i) B2C contract; (ii) B2B contract; (iii) Consent-based B2C and B2B contract. While in block-2, we show the CCV with the fourth scenario (i.e., automatic detection of contract breaches based on informed consent, where the consent has expired and the contracts—based on that consent—are still running). In smashHit, a B2C contract is created between an insurer (acts as a data subject) and LexisNexis (acts as a data controller). While a B2B contract is made between LexisNexis (acts as a data controller) and an organisation (as a data processor). To make a clear distinction among all four scenarios, we assigned different colours to scenarios. In block-1, the B2C contract scenario is presented in baby blue colour, the B2B contract scenario is in iceberg colour, and the informed-consent base contract scenario is in fresh air colour. Alice blue colour is assigned to present the fourth scenario of the CCV in block-2. In the following subsections, we present each of them in more detail.

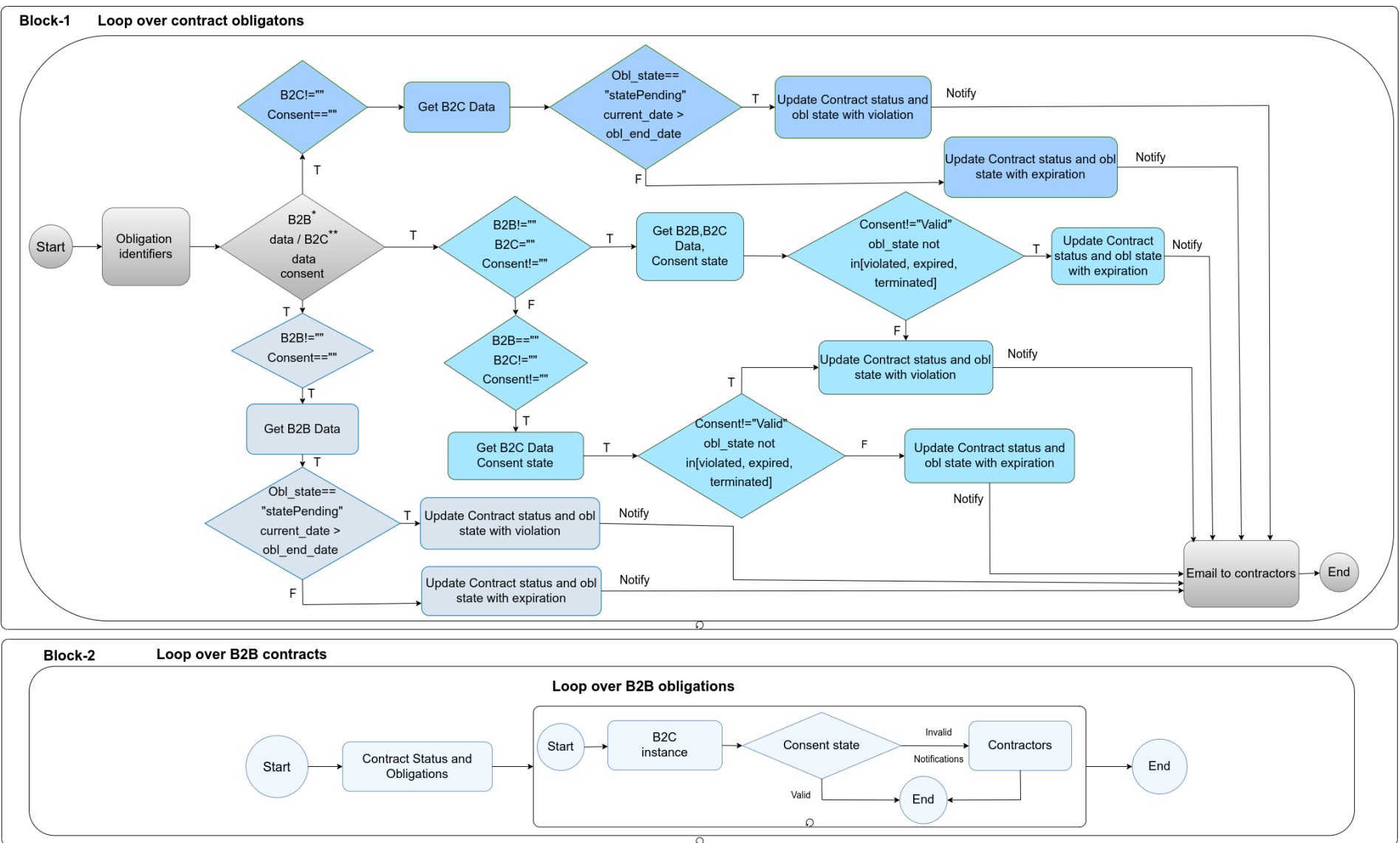

**Figure 4.** CCV process overview.

### 3.3.1. B2C or B2B Contracts

The first two scenarios are very similar, with the difference being in the contract's category. Both must satisfy the following conditions for compliance verification: (i) a B2B or B2C contract and (ii) the consent information must be empty. If the condition result is evaluated as *true*, contractual information of the B2B or B2C is extracted from the KGs via the SPARQL endpoint. Based on this information, another conditional check is performed on the contractual clause with its states and end date. If this is the case, the condition result is true, and the contract status and its clause states are updated in the KGs with violation information. Otherwise, both contract status and clause states update with expiration information. In both cases, the tool sends notifications automatically to contractual parties with violation or expiration information.

### 3.3.2. Consent-Based Compliance Verification on B2C Contracts

The third scenario is based on consent, which has two parts: B2C contract and B2B contract. In the first part, the conditions (*B2B==""* and *B2C!=""* and *Consent!=""*) must be true to perform compliance verification checks on B2C contracts. Once the condition is *true*, the tool extracts B2C contract information from the KGs. Based on this information, the tool validates it with the consent state and the state of the contractual clause. If the condition result is *true*, the tool updates the contract's status and the contractual clause states with violation information. In a case where the condition's result is *false*, the contract status and contractual clause states will update with expiration information. The tool notifies the contractual parties automatically based on this violation or expiration information. This process repeats for the second part, which is B2B contract compliance verification. In a case where the consent state is invalid, it must also expire all contracts associated with that consent. This process repeats and executes until there is no clause left for validation.

### 3.3.3. Consent-Based Compliance Verification on B2B Contracts

The tool also supports detecting contract violations based on consent automatically. To illustrate it, we consider a data subject (e.g., a person and software) possessing a B2C contract with LexisNexis (i.e., the data controller) and consenting to share data for five months. Based on this contract, LexisNexis makes a B2B contract with an organisation ABC (i.e., data processor) to obtain benefits for selling the data. For illustration purposes, let us assume that the data subject revoked consent after two months. The contracts associated with that consent must be terminated or expire. Since consent has been revoked, the B2C contract must also be terminated or must expire. The B2B contract, which is created based on a B2C contract, must also be terminated or expire. Let us assume for any reason that the B2B contract is still running. In that case, how will the data subject know about this contract breach? Our tool supports detecting this type of breach automatically and updates the data subjects about each contract associated with them. We present the process of this scenario in block-2 of Figure 4.

In the process of Block-2, we extract all B2B contracts with their contractual clause information. To create a consent-based B2B contract, we have a B2B contract clause possessing a B2C contract reference. With the help of this reference, we can extract the consent state from a B2C contract. Furthermore, we perform a compliance check with the consent state and contract status. If the consent state is invalid and the contract status has not ended or expired, then the tool sends notifications to the data subjects about this violation.

## 4. Architectural Design and Implementation

This section details the architectural design of the CCV, which is presented in Section 4.1 and its implementation details are presented in Section 4.2.

### 4.1. CCV Architectural Design

Figure 5 presents the CCV architectural design. It follows a micro-services architecture pattern. A micro-services architecture pattern is one in which all modules are cohesive,

independent processes that interact through messages [48]. The Service Layer is a key component, which comprises the Core, the Resources, the Application Programming Interface (API) Layer, and the contract compliance scheduler. We describe each of them in the following subsections.

### 4.1.1. Core

The Core module is divided into two sub-modules: Data Processing and Shared Services. The first one is responsible for data management to support required operations, such as contract creation, auditing or controlling, and compliance-verification checking. The query processor and storage are two sub-components for supporting data processing operations. The query processor component contains the SPARQL queries required to deal with contracts (e.g., contract data in KGs), while the storage module handles the query processor's execution. Shared services include modules that assist other modules in their operations, such as contract and compliance verification checks.

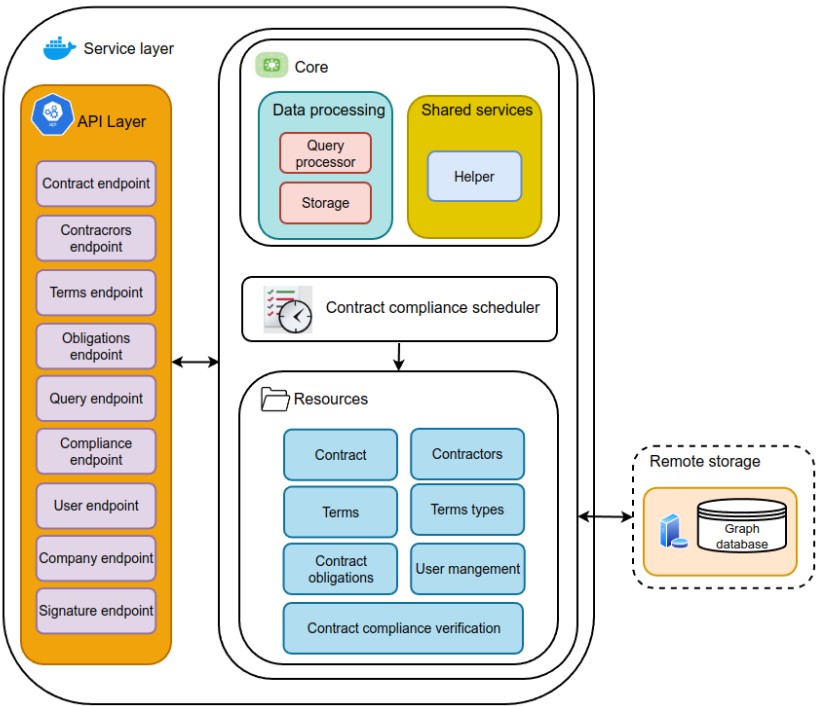

**Figure 5.** Architectural design of the automated GDPR compliance verification tool.

### 4.1.2. API Layer

The API layer is used to interact with the CCV tool. It provides access to the compliance verification tool's functionalities via REST (REpresentational State Transfer) endpoints. Contract search, contractual parties management, contract audit, and contract compliance are the core features of the API layer.

### 4.1.3. Resources

The Resources component is a part of the Service Layer, which contains classes, such as Contract, Contractual Parties, Contractual Terms, Contractual Clause Types, Contractual Clause, and Contract Compliance. These are required for the management and compliance verification of digital contracts. Each class has Create, Read, Update, and Delete (CRUD) operations. Search by ID (e.g., contract id, contractual party's id, and clause id) and searching the bulk of records are two common types of search. The contract compliance component is responsible for performing automated compliance verification checks. Figure 6 presents the JSON (JavaScript Object Notation) schema and the semantic representation of a B2B contract.

Figure 6 represents an overview of a contract instance from our knowledge graph and all information related to it. The centre node (in red colour) represents an instance of the class *fibo-fnd-agr-ctr:Contract*. This instance's label has been encrypted for security and privacy reasons. All other nodes represent entities related to that specific contract instance. For example, a contract can have several contractors associated with it (see the nodes connected to the contract instance via the "has contractors" relationship).

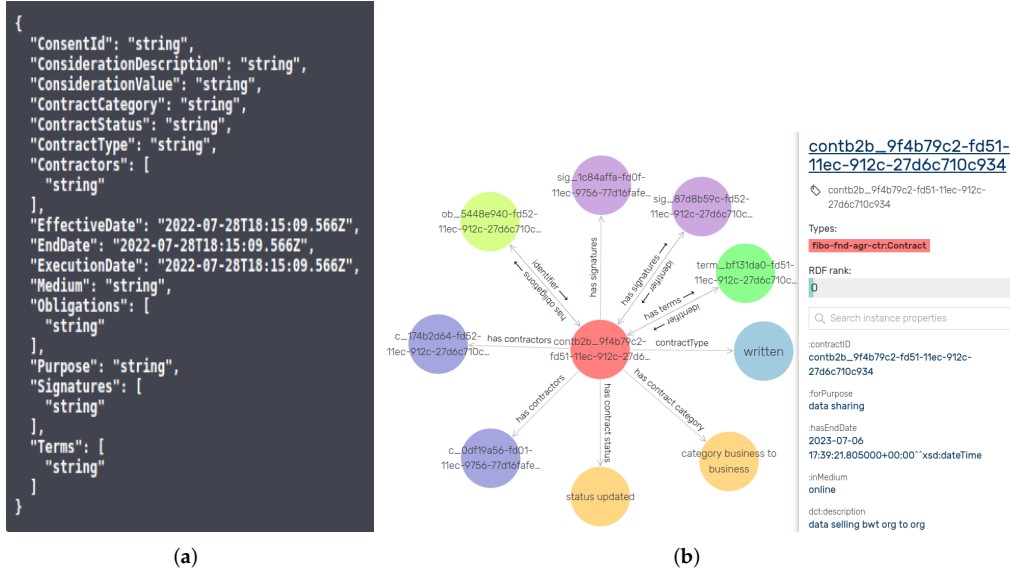

(**a**)                                              (**b**)

**Figure 6.** A snapshot of the JSON schema used for contract and the contract module's creation (or representation) of contract in the legal KG. (**a**) Contract JSON schema. (**b**) KG representation of a contract in GraphDB.

### 4.1.4. Remote Storage

For the construction and management of contracts, Ontotext GraphDB [49] is used, which supports the RDF and SPARQL. The SPARQL endpoint is used to perform CRUD operations on digital contracts, while RDF is used in the construction of semantic models of contracts.

### 4.1.5. Contract Compliance Scheduler

The Flask APScheduler [50] is used to handle time-based job scheduling tasks for automated detection of contract breaches. For instance, we set up a scheduling task for contract breach detection, which executes every day at 01:00 a.m. The data controller makes compliance verification checks directly by calling the contract compliance endpoint through the contract's REST APIs endpoints. The source code of the entire process can be found on GitHub [51].

### 4.1.6. Contract REST API

For interactions with the tool, the API Layer implements REST endpoints. For an ideal representation of the API documentation, swagger [52] is used. The swagger REST APIs endpoints for contracts can be found on GitHub [51], which requires performing CCV checks and managing digital contracts. We divide the contract REST APIs endpoints into seven parts: contracts; contractual parties; clause types; contractual terms; contract contractors (contractual parties, which are associated with a particular contract); contractual clauses; contract signatures; and contract compliance. Each part in terms of functionality is able to perform CRUD operations. For binding requests from swagger API for KG, custom contract schema are used as shown in Figure 6. This contract schema comprises basic information (e.g., contract category, contract types, and purpose) and collections of contractual parties, contractual terms, contractor signatures, and contractual clauses.

*4.2. Implementation*

This section details the tool's implementation based on the use cases and industrial requirements discussed in Section 1. The implementation of TOMs is presented in Section 4.2.1. We describe the libraries used for this implementation in Section 4.2.2, while in Section 4.2.3, we present the implementation details for each component of the tool.

### 4.2.1. The Implementation of TOMs

Performing the CCV with GDPR, our solution follows the "data protection by design and by default" principle. For this principle, implementing TOMs is a key requirement according to GDPR (Art. 25 (1)). The adoption of internal policies (Rec. 78) states that it is the responsibility of the data controller (or data processor) to implement TOMs, ensuring that processing is performed under GDPR (Art. 4 (7), Art. 24). Our tool implements the following TOMs.

#### Data Encryption

The first TOM relates to confidentiality (Art. 32 (1) (a)) to encrypt the processing data. For encryption, we use the deterministic searchable encryption technique. Two algorithms the Rivest–Shamir–Adleman (RSA) [53] with Public Key Cryptography (PKCS) Standards and asymmetric Advanced Encryption Standard (AES) [54] are used for this purpose. Further, by implementing authentication procedures and identity management, only registered components have access to endpoints.

#### Protection against External Influences on Systems and Services

This TOM relates to Art. 32 (1) (b), which is defined as "the ability to ensure the ongoing confidentiality, integrity, availability, and resilience of processing systems and services" to ensure that the systems and services are planned correctly and according to the intended purpose. The tool implements security measures (e.g., authentication procedures see detail in Section 4.2.3) and user-based access to prevent unauthorised data access.

#### Documentation of Data Syntax

The third TOM relates to the documentation of data, its availability and resilience (Art. 32 (1) (b)). The entire code follows the Python Enhancement Proposals (PEP)-8 [55] coding convention and is commented for better understandability. For an ideal representation of the API documentation, Swagger [52] is used.

#### Reduction in Non-Required Attributes of Data Subjects

Our fourth TOM is used to enable data minimisation according to GDPR (Art. 32 (1) (d), 25 (1)). Our tool implements the data minimisation requirements according to GDPR by establishing retention periods (e.g., dates) for personal data processing to ensure GDPR contract compliance verification. For example, our contract's REST APIs endpoint creation defines a minimal set of variables, such as purpose and dates (execution date and end date for retaining the data only for as long as it is necessary to fulfil the purpose of processing).

#### Role Concepts with Graduated Access Rights Based on Identity Management and a Secure Authentication Process

This TOM relates to the purpose of limitation according to (Art. 32 (1) (d), 25 (1))), which is about testing, assessing and evaluating the effectiveness of TOM to ensure the security of data processing. It takes the purpose of the limitation into account and defines permissible purpose changes. Our tool uses the JavaScript Object Notation (JSON) and Web Tokens (JWT) [56] based access control. Furthermore, user-based access on endpoints is implemented.

Translating legal requirements into technical implementations is not easy. The Standard Data Protection Mode (SDM) [57] provides appropriate measures, transforming the GDPR legal bases to qualify for TOMs. A summary of GDPR requirements mapped with

their data protection goals is described in [57] (Table in Section C2, p. 28). The SDM is as follows:

1. Systematises data protection requirements in the form of protection goals;
2. Systematically derives generic measures from protection goals, supplemented by a catalogue of reference measures;
3. Systematises the identification of risks in order to determine protection requirements of the data subjects resulting from the processing; and
4. Offers a procedure model for modelling, implementation, and continuous control and testing of processing activities.

### 4.2.2. System Setup for Evaluation

We summarise the libraries and software that were used in this implementation in Table 1. We selected these libraries because of our tool's requirements. For instance, GraphDB was selected due to having capabilities, such as more intuitive data visualisation, storage, and management. The free edition of GraphDB is not sufficient for simultaneous queries because it does not support concurrency or parallelism of more than two queries. In order to alleviate this issue, the Enterprise Edition (EE) of GraphDB can be deployed instead. A Docker container in a system with 32 GB (gigabyte) random-access memory (RAM), a 1.7 gigahertz (GHz) AMD Ryzen 7 PRO 4750U processor, and 1 terabyte (TB) storage is used for deploying the service layer. Linux with variant distributions, such as Ubuntu and Debian, is used for all deployment setups.

**Table 1.** List of software (or libraries) that were used in the implementation.

| Software (or Libraries) | Version |
|:---:|:---:|
| Python [58] | 3.8 |
| Flask [59] | 1.1.2 |
| Flask-RESTful [60] | 0.3.8 |
| Flask-SQLAlchemy [61] | 2.5.1 |
| Python Requests | 2.25.1 |
| Flask Apispec [62] | 0.11.0 |
| Pycryptodome [63] | 3.10.1 |
| SPARQLWrapper [64] | 1.8.5 |
| Docker ([65] Community Edition) | 20.X |
| SQLite | 2.6 |
| GraphDB free edition [49] | 9.4.1 |
| Protégé | 5.5.0 |
| Pyjwt [66] | 1.7.1 |

### 4.2.3. Automated GDPR CCV Tool Implementation

This section provides the implementation details of each component of the CCV tool, such as the API layer and CCV layers.

#### API Layer

The main functionality of the API layer is to implement the REST endpoints for contracts. It enables user-based access as demanded by GDPR's integrity and confidentiality principle (Art. 5 (1) (f)) by custom JWT implementation. All the contract's REST API endpoints are only accessed through a valid JWT token, which is created upon successful login. Furthermore, the standard REST practices, such as OpenAPI Specification (OAS) version 2.0 and swagger, are used in implementing the API layer for describing the contract's REST endpoints (see Figure 7).

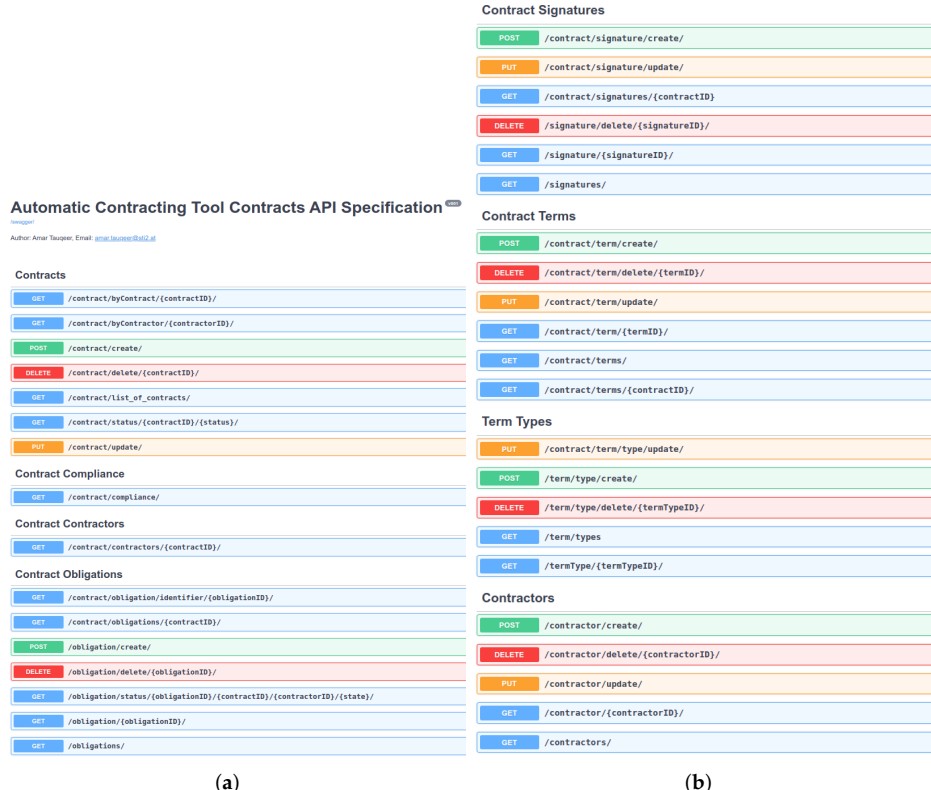

**Figure 7.** Contract REST API's endpoints in Swagger. (**a**) Part 1. (**b**) Part 2.

Data Processing

The data processing module comprises predefined SPARQL queries with the contractual information to be filled in during the run-time. These queries are organised based on the Resource (see detail in Section 4.1.3) component of the CCV tool. In Figure 8, we present a snapshot of the SPARQL query, which is used to extract all information with respect to a contract such as contract contractors, terms, obligations, and contract category. It contains two functions *prefix* and *get_all_contractors*. All namespaces are stored in the first one required for the execution of the SPARQL query, while the second one stores the SPARQL query.

```python
def prefix(self):
    prefix = textwrap.dedent("""PREFIX : <http://ontologies.atb-bremen.de/smashHitCore#>
        PREFIX rdf: <http://www.w3.org/1999/02/22-rdf-syntax-ns#>
        PREFIX dc: <http://purl.org/dc/elements/1.1/>
        PREFIX dpv: <http://www.w3.org/ns/dpv#>
        PREFIX prov: <http://www.w3.org/ns/prov#>
        PREFIX dcat: <http://www.w3.org/ns/dcat#>
        PREFIX fibo-fnd-agr-ctr: <https://spec.edmcouncil.org/fibo/ontology/FND/Agreements/Contracts/>
        PREFIX dct: <http://purl.org/dc/terms/>
    """)
    return prefix

def get_all_contracts(self):
    query = textwrap.dedent("""{0}
        select *
        where{{
        ?Contract rdf:type fibo-fnd-agr-ctr:Contract;
            :contractID ?contractId;
            :hasContractStatus ?contractStatus;
            :hasContractCategory ?contractCategory;
            dct:identifier ?consentId;
            :forPurpose ?purpose;
            :contractType ?contractType;
            fibo-fnd-agr-ctr:hasEffectiveDate ?effectiveDate;
            fibo-fnd-agr-ctr:hasExecutionDate ?executionDate;
            :hasEndDate ?endDate;
            :inMedium ?medium;
            dct:description ?consideration;
            rdf:value ?value .
        }}
    """).format(self.prefix())
    return query
```

**Figure 8.** A snippet of code from the query processor module.

Shared Service

Two functions *function_map* and *list_to_query* are implemented by the shared services module as shown in Figure 9. The other modules, such as data processing and compliance verification can use these shared services. The *list_to_query* function is used to convert the array of JSON inputs into the SPARQL query format for supporting contract creation activities by the contract module, while the *function_map* is used to perform the mapping to the actual function.

```python
def function_map(self, name):
    """ Map to actual function
    :param name: name which function to map
    :return: function name
    """
    mapfunc = {
        "get_all_contracts": self.get_all_contracts,
        "get_contract_by_contractor": self.get_contract_by_contractor,
        "get_contract_by_provider": self.get_contract_by_provider,
        "get_contract_by_id": self.get_contract_by_id,
        "get_signature_by_id": self.get_signature_by_id,
        "get_contractor_by_id": self.get_contractor_by_id,
        "get_company_by_id": self.get_company_by_id,
        "get_all_contractors": self.get_all_contractors,
        "get_all_companies": self.get_all_companies,
        "get_all_terms": self.get_all_terms,
        "get_all_signatures": self.get_all_signatures,
        "get_contract_signatures": self.get_contract_signatures,
        "get_term_type_by_id": self.get_term_type_by_id,
        "get_term_by_id": self.get_term_by_id,
        "get_obligation_by_id": self.get_obligation_by_id,
        "get_all_obligations": self.get_all_obligations,
        "get_contract_obligations": self.get_contract_obligations,
        "get_all_term_types": self.get_all_term_types,
        "get_contract_terms": self.get_contract_terms,
        "get_contract_contractors": self.get_contract_contractors,
        "get_contract_compliance": self.get_contract_compliance,
        "contract_update_status": self.contract_update_status,
        "get_obligation_identifier_by_id": self.get_obligation_identifier_by_id,
        "get_signature_identifier_by_id": self.get_signature_identifier_by_id,

    }
    return mapfunc[name]

def list_to_query(self, data, whatfor):
    """ Convert list to query
    :input: list
    :returns: SPARQL query string
    """
    querydata = ""
    for vlaue in data:
        strs = ":" + whatfor + " :" + vlaue + ";\n"
        querydata = strs + querydata
    return querydata
```

**Figure 9.** A snippet of code from the helper module.

Contract Compliance Scheduler

The Flask APScheduler [62] is used to handle time-based job scheduling tasks for automated detection of contract breaches. For example, the tool sets up a scheduling task for contract breach detection, which executes every day at 01:00 a.m. Furthermore, the data controller makes a compliance verification check directly by calling the contract compliance endpoint through the contract's REST API endpoint. Figure 10 presents a compliance verification scheduling task based on the current date.

```python
def compliance():
    CONTRACT_URL = "https://actool.contract.sti2.at/contract/compliance/"
    data = requests.get(CONTRACT_URL)
    data = data.json()

if __name__ == '__main__':
    scheduler.add_job(id='Contract compliance task', func=compliance, trigger='interval', minutes=1440)
    if current_date >= date(some date):
        scheduler.start()
```

**Figure 10.** A snippet of code for scheduling the compliance verification check.

Resources

The Resource component of CCV implements sub-modules, such as Contract, Contractual Parties, Contractual Terms, Contractual Clause Types, Contractual Clause, and Contract Compliance, which are used for the management and to perform compliance verification checks on contracts. Each class has its procedures for performing CRUD operations. For instance, creating a new contract requires contract data in JSON format following JSON schema, as shown in Figure 7. This scheme is used to transform the contract data into KG and is validated with marshmallow [67]. Marshmallow is a framework-agnostic library for converting complex data types, such as objects, to and from native Python data types.

Each component performs the following similar functionalities: (i) extracting all details (records) of the component; (ii) extracting component specific details (based on IDs e.g., contractID, contractorID); (iii) component creation; (iv) updating a particular component; and (v) deleting a specific component. All components of the Resource module perform partial and full auditing. Figure 11 presents a snapshot of a contract's partial and full audit in the JSON Schema. The basic information of the contract, such as contract category and contract dates (e.g., start, effective, and execution), is provided for the partial contract audit, as shown in Figure 11. While Figure 11 presents a full contract audit information in JSON Schema. It contains not only the basic information of the contract but also other contractual information, such as a collection of contractual parties, a collection of contractual terms, and a collection of contractual clauses.

```
{                                                     {
  "ContractId": "contb2c_001",                          "ContractId": "contb2c_001",
  "ConsentId": "const_001",                             "ConsentId": "const_001",
  "ConsiderationDescription": "data sharing",           "ConsiderationDescription": "data sharing",
  "ConsiderationValue": "2000",                         "ConsiderationValue": "2000",
  "ContractCategory": "categoryBusinessToConsumer",     "ContractCategory": "categoryBusinessToConsumer",
  "ContractStatus": "statusCreated",                    "ContractStatus": "statusCreated",
  "ContractType": "wirtten",                            "ContractType": "wirtten",
  "Contractors": [                                      "Contractors": [
    "c_004","c_005"                                       "c_001","c_002"
  ],                                                    ],
  "EffectiveDate": "2022-07-13T10:32:16.088Z",          "EffectiveDate": "2022-07-13T10:32:16.088Z",
  "EndDate": "2023-07-13T10:32:16.088Z",                "EndDate": "2023-07-13T10:32:16.088Z",
  "ExecutionDate": "2022-07-13T10:32:16.089Z",          "ExecutionDate": "2022-07-13T10:32:16.089Z",
  "Medium": "online",                                   "Medium": "online",
  "Obligations": [                                      "Obligations": [
    "ob_004"                                              "ob_001","ob_002"
  ],                                                    ],
  "Purpose": "data selling",                            "Purpose": "data selling",
  "Signatures": [                                       "Signatures": [
    "s_004","s_005"                                       "s_001","s_002"
  ],                                                    ],
  "Terms": [                                            "Terms": [
    "term_type_001"                                       "term_type_001"
  ]                                                     ]
}                                                     }
                    (a)                                                   (b)
```

**Figure 11.** A snapshot of contract partial and full audit JSON Schema. (**a**) Partial audit. (**b**) Full audit.

The CCV (see detail in Section 3.2) implements an automated compliance verification check to perform on contracts. This compliance check performs only on active contracts (i.e., a contract having status, such as created, pending, and updated). The CCV implementation is based on four scenarios discussed in Section 3.3. The implementation in the first two scenarios is based on B2C and B2B contracts. The third scenario is based on consent-based contracts. The fourth scenario performs the compliance checks on B2B contracts, where the consent has expired but the contracts (associated with that consent) are still running. Each component's implementation details with respect to the Resource module can be found on GitHub [51].

Security

Two algorithms RSA [53] and AES [54] are used to ensure secure data processing in the CCV tool. The RSA algorithm's proven capability and security robustness over the last 30 years is a valid reason for its selection. While considering the de facto standard for symmetric encryption and standardised by the National Institute of Standards and

Technology (NIST) as an encryption technique, the AES algorithm is selected, which is fast and secure [68]. The function *key_generate* is used to create and export the public and private keys using RSA. For data encryption and decryption, the security module implements two functions *rsa_aes_encrypt* and *rsa_aes_decrypt*. The Public-Key Cryptography Standards (PKCS) # 1 OPTIMAL ASYMMETRIC ENCRYPTION PADDING (OAEP) [69] padding scheme is used by the RSA's implementation, which is defined by RFC 8017. To encrypt and decrypt the keys for symmetric encryption algorithms, RSA is used. The complete implementation details can be found on GitHub [51].

## 5. Evaluation

This section presents the evaluation of our tool with a focus on performing CCV compliance functionalities. It is based on tools' key functionalities, such as contract creation, contract audit, and CCV checks. Both use cases (UC1 and UC2) require scalable solutions to handle end users. In Section 5.1, we present the CCV performance evaluation, while we show the TOMs evaluation in Section 5.2. Furthermore, for CCV implementation and performance evaluation, we use the system's setup, as described in Section 4.2.2.

### 5.1. CCV Performance Evaluation

To evaluate the CCV performance, we created ten different contract instances based on UC1 and UC2 and measured their execution times. The process repeats to create instances of contract terms and contractual clauses. The contract creation process is divided into five parts: (i) contract's basic information, (ii) contractors, (iii) contract terms, (iv) contractor signatures, and (v) contractual clauses. The execution time of a contract creation is based on the total execution time of all the above five parts. For this performance evaluation, the contract's information is provided manually. For this performance evaluation, the information about the instances of contract, contractual terms, contractual clauses, and terms types can be found on GitHub [51] (see contract-creation.ods file in the evaluation folder).

Before discussing evaluation results, we introduced terms, such as *contract create instance (CTI)*, *contract audit (CTA)*, *contract terms create (CTT)*, *contract terms audit (CTTA)*, *contract obligation create (CTO)*, *contract obligation audit (CTOA)*, and *COMP* for the instances of contract creation, contract audit, contract terms, contract obligations, and contract compliance. Figure 12 represents the contract creation (including five parts) instances (*CTI1*, *CTI2*, . . . *CTI10*) on the x-axis, while the execution time (in minutes) of each contract creation instance shows on the y-axis. On the right side of Figure 12, we have contract audit instances (*CTA1*, *CTA2*, . . . *CTA10*) on the x-axis and execution time (in minutes) on the y-axis in the Figure 12. Similarly, in Figure 13, we can see the evaluation results of contract terms create and audit with their instances. In addition, Figure 14 represents the evaluation results of contractual clauses creation and audit with their instances, where instances are shown on the x-axis and execution time on the y-axis. Finally, we present the compliance verification results in terms of execution time in Figure 15, where the instances (*COMP1*, *COMP2*, . . . *COMP10*) are presented on the x-axis and the execution time (in seconds) on the y-axis.

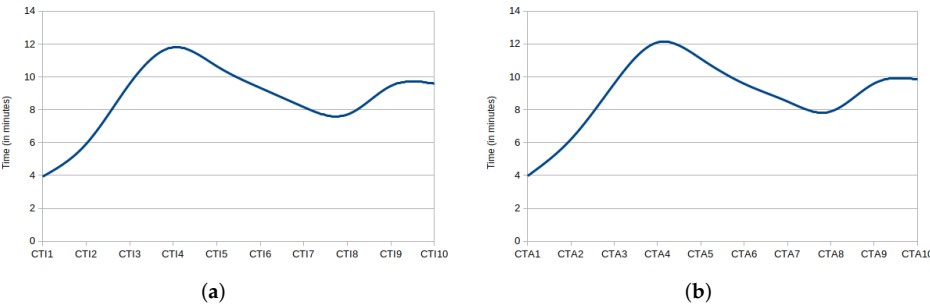

(a)　　　　　　　　　　　　(b)

**Figure 12.** Performance evaluation on contract creation and audit. (**a**) Time spent on contract creation. (**b**) Time spent on contract audit.

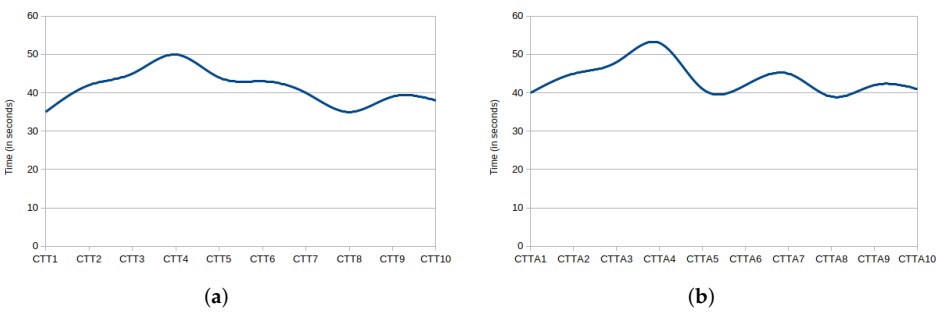

(a)　　　　　　　　　　　　(b)

**Figure 13.** Performance evaluation on contract term creation and audit. (**a**) Time spent on contract term creation. (**b**) Time spent on contract term audit.

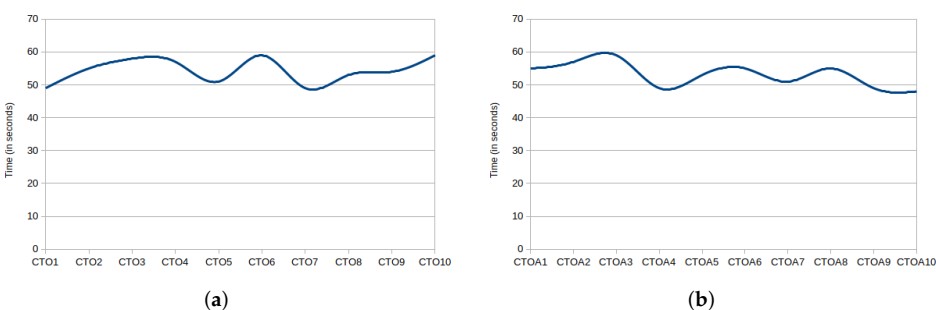

(a)　　　　　　　　　　　　(b)

**Figure 14.** Performance evaluation on contract clause creation and audit. (**a**) Time spent on contractual clause create. (**b**) Time spent on contractual clause audit.

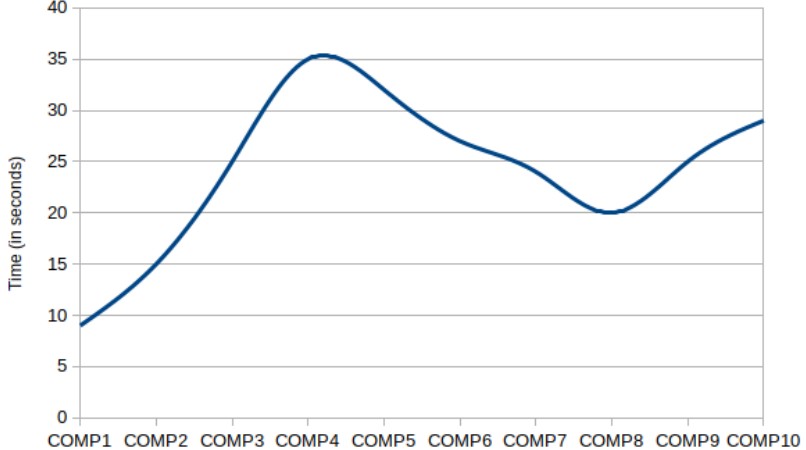

**Figure 15.** Time spent on CCV.

The minimum time spent on a contract creation process is 3.55 min, while 11.48 is the maximum time spent on a contract as shown in Figure 12. The *CTI4* took 11.48 min because it has two contract terms and four contractual clauses, whereas *CTI1* only has a contract term and a contractual clause. Similarly, the maximum time of 4.50 min was spent on contractual clauses (four clauses), 2.50 min on contract (two terms), 0.42 s on contractor signatures (three signatures), 2.00 min on contractual parties (two contractors), and 1.40 min on the contract's basic information. Contract audit and creation processes have taken almost the same execution time because both have the same contents as shown in Figure 12a,b. We did not consider partial creation and audits here.

We also measured the performance evaluation on the contract's term and contractual clause, which are shown in Figures 13a,b and 14a,b. The average time spent on the contract term creation or audit is 0.40 s, as shown in Figure 13a,b, while the contractual clauses creation or audit took an average of 0.55 s, as shown in Figure 14a,b. Based on contract creation instances depicted in Figure 12, we measured the time spent over compliance verification in Figure 15. It shows a strong relationship with contract instances, and if the instances take more time, the compliance verification will also take more time. For example, the *COMP4* (compliance instance related to *CTI4*) took 36 s to complete, whereas *COMP8* (related to *CTI8*) took only 20 s. More information about the contract evaluation performance is shown in Table 2 (the fastest and slowest measurements in terms of time are highlighted in bold). The contract contents (e.g., terms, and obligations) can also affect the execution time of the contract creation, audit, and compliance. The encryption and decryption of the information can result in higher time in performance evaluation. However, these are also required to increase security measures. These extra time-consuming activities are associated with compliance verification and cause the CCV tool to slow down.

**Table 2.** Contract performance evaluation.

| ID | Contract Basic In-formation (Time in Minutes) | Contractual Parties (Time in Minutes) | Contractual Term (Time in Minutes) | Contractual Clauses (Time in Minutes) | Contractor Signatures (Time in Minutes) | Total (Time in Minutes) |
|----|----|----|----|----|----|----|
| 1 | **1.00** | **0:50** | 0.16 | 0.44 | 0.16 | **3.55** |
| 2 | 1.05 | 1:55 | 0.32 | 1.58 | 0.30 | 5.54 |
| 3 | 1.20 | **2:00** | 1.50 | 3.50 | 0.35 | 9.34 |
| 4 | 1.30 | 1:58 | **2.50** | **4.50** | 0.40 | **11.48** |
| 5 | **1.40** | **2:00** | 1.50 | 3.52 | 0.37 | 10.39 |
| 6 | 1.20 | 1:57 | 1.48 | 3.40 | 0.35 | 9.33 |
| 7 | 1.30 | **2:00** | 1.20 | 2.40 | 0.40 | 8.16 |
| 8 | 1.10 | 1:58 | 1.30 | 2.30 | 0.35 | 7.42 |
| 9 | 1.25 | **2:00** | 1.40 | 3.45 | 0.37 | 9.45 |
| 10 | 1.35 | 1:55 | 1.58 | 3.25 | **0.42** | 9.34 |

To evaluate the correctness of the CCV tool, we performed unit tests with 28 different test case scenarios. The evaluated test cases include the CRUD operations relating to contracts, such as contract terms and contractual obligations. Moreover, the test cases also include the CCV tool's compliance verification operations. The CCV compliance verification unit test cases include 5 different test scenarios described in Section 3.3. Figure 16 shows a code snippet of the unit test for the B2B contract scenario without consent. As shown in Figure 16, the test case takes the contract's ID, status, current date, obligation state, obligation end date, and obligation ID for compliance verification. Further, a condition (i.e., *current date > obligation end date and obligation state = 'Pending' and b2b contract status not in ('Violated', 'Terminated', 'Expired')*) is checked. The contract status and obligation state must be updated by the tool if the condition result yields true.

```python
# handle single business to business contract without consent
def test_b2b_without_consent(self):
    current_date = "2023-09-06"

    b2b_contract = "contb2b_e45dc546-2e9e-11ed-be7d-3f8589292a29"
    b2b_contract_status = "statusCreated"

    obligation_state = "statePending"
    obligation_end_date = "2023-07-06"
    obligation_id = "ob_1e5293a0-2e98-11ed-be7d-3f8589292a29"

    if current_date > obligation_end_date and obligation_state == 'statePending' and b2b_contract_status not in (
            'statusViolated', 'statusTerminated', 'statusExpired'):
        expected_status = "statusViolated"
        expected_obligation_state = "stateViolated"

        r = requests.get(ContractApiTest.CONTRACT_URL +
                        "/status/{}/{}/".format(b2b_contract, expected_status))
        self.assertEqual(r.status_code, 200)

        r = requests.get(ContractApiTest.CONTRACT_URL +
                        "/obligation/states/{}/{}/".format(obligation_id, expected_obligation_state))
        self.assertEqual(r.status_code, 200)
```

**Figure 16.** A unit test code snippet for B2B test scenario to check the CCV correctness.

The unit test cases with their results logs, and the source code are available on GitHub [51]. Our evaluation of unit test cases demonstrates that the CCV tool performs the intended tasks, such as compliance verification and contract creation correctly.

*5.2. TOMs Evaluation*

In Section 4.2.1, we summarised and evaluated five data protection goals associated with TOM regarding GDPR (data processing) for contracts. We evaluated the contracts module manually. We discussed the manual evaluation in Section 5.1. For instance, the contractual parties' personal data are encrypted first and then stored in the KG, which is related to GDPR (Art. 32 (1) (a)) confidentiality (see detail in Section 4.2.1). Similarly, the PEP-8 coding convention is used for the documentation of data syntax. For the automated procedure, we wrote test use cases and executed them using Python's unit test framework [70]. We have different conditions to make these tests. For example, testing the endpoint *GetContractContractor*, we provide the contract number. In the case of providing the contract number, the test case returns a success response. The test case returns a failed response in case of missing the contract number. More information about each test case can be found on GitHub [51].

## 6. Conclusions and Future Work

By building on our previous work in [24], in this paper, we presented our CCV (Contract Compliance Verification) tool for digital contract management based on knowledge graphs. To be specific, we presented an approach for automated CCV checks over digital contracts. Further, we discussed factors that must consider the technical requirements to satisfy industry requirements as discussed in Section 1. The CCV tool has micro-services architecture and utilises an ontology and a knowledge graph, which support the interoperability of data.

The CCV tool supports contract management, based on consent, with B2C (Business-to-Consumer) and B2B (Business-to-Business) contracts that can be generalised to other domains. Our tool supports not only consent-based contract generation but also considers scenarios in which consent is not required (see Section 4). We evaluated the CCV tool with the performance and scalability regarding contracts. We also evaluated the CCV methods correctness by performing unit test cases. This research is conducted in collaboration with legal experts and industrial partners. It can help SMEs (small and medium enterprises) in binding GDPR (General Data Protection Regulation) legal bases with data sharing contracts. The CCV tool improves both the compliance verification process and contract lifecycle. Future studies include (i) improving the signing process with digital signatures; (ii) implementing digital licensing on contracts using DALICC [71]/Licence Clearance Tool (LCT) [72]; (iii) improving the negotiation process where the data subject will have more options to collaborate on making contract clauses; (iv) performing validation to graph-based data using Shapes And Constraints Language (SHACL); (v) extracting the existing contracts (e.g., paper contracts and unstructured contracts) from external data sources to

translate into RDF (Resource Description Framework); and (vi) optimising the performance of the tool.

**Author Contributions:** Conceptualization, A.T. and A.K.; methodology, A.T.; software, A.T. and T.R.C; validation, A.T., T.R.C. and A.A.; formal analysis, A.T. and A.K.; investigation, A.T., A.A. and T.R.C; resources, A.T.; data curation, A.T.; writing—original draft preparation, A.T.; writing —review and editing, A.T., A.K., A.A., T.R.C. and A.F.; visualization, A.T.; supervision, A.F.; project administration, A.F.; funding acquisition, A.F. All authors have read and agreed to the published version of the manuscript.

**Funding:** This work is supported by the Horizon 2020 project smashHit (grant number 871477).

**Data Availability Statement:** Not applicable.

**Acknowledgments:** We express our gratitude to Friederike Knoke and Samuel Iheanyi Nwankwo from Leibniz Universität Hannover (LUH) Institut für Rechtsinformatik (IRI) for the legal analysis of the contracts for our use cases. Further, we thank our industry collaborators LexisNexis Risk Solutions, Volkswagen AG, Infotripla and Forum Virium Helsinki for supporting the use cases for our work. We also thank our colleagues Rainer Hilscher and Antonio J. Roa-Valverde from University of Innsbruck (UIBK) for participating in the discussions and providing appropriate feedback on our work.

**Conflicts of Interest:** The authors declare no conflict of interest. The funders had no role in the design of the study; in the collection, analyses, or interpretation of data; in the writing of the manuscript, or in the decision to publish the results.

## Appendix A

*Semantic Models Prefix*

```
@prefix rdf: <http://www.w3.org/1999/02/22-rdf-syntax-ns#> .
@prefix rdfs: <http://www.w3.org/2000/01/rdf-schema#> .
@prefix owl: <http://www.w3.org/2002/07/owl#> .
@prefix dct: <http://purl.org/dc/terms/> .
@prefix prov: <http://www.w3.org/ns/prov#> .
@prefix gconsent: <https://w3id.org/GConsent#> .
@prefix dpv: <http://www.w3.org/ns/dpv#> .
@prefix fibo-fnd-agr-ctr:
<https://spec.edmcouncil.org/fibo/ontology/FND/Agreements/Contracts/> .
@prefix smashHitCore: <http://ontologies.atb-bremen.de/smashHitCore#> .

@prefix dcat: <http://www.w3.org/ns/dcat#> .
@prefix time: <http://www.w3.org/2006/time#> .
@prefix LC: <https://www.omg.org/spec/LCC/Countries/CountryRepresentation/> .
```

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
