# Peer review of "Automated GDPR Contract Compliance Verification Using Knowledge Graphs"

_information, doi:10.3390/info13100447_

Round 1

Reviewer 1 Report

The author presents a KG-based solution for digital contracting which can perform the CCV checks on contracts and bind GDPR with data sharing contracts. Semantic Modelling is used for B2B and B2C. It is an innovative idea. Although the paper is well-written and with good organization, there are some issues should be discussed:

1) The author uses some abbreviations of proper noun, which first appear, such as GDPR, TOMs and ACT. These abbreviations without detailed interpretation are unfriendly for some reader in non-professional domain.

CCV(contract compliance verification) is mentioned many times in this paper, but some CVV also appeared (e.g. first line in 3.1). There is no explict definition of CVV, the author should give a detailed explanation.

2) In section 2, the author introduces some related work in the construction and management of contracts. However, it just enumerates some related methods, and the analysis about key technology, advantages and disadvantages of these methods should be given in further.

3) The figure 1 just shows that unavoidable process which is contract management(e.g. Contract Request/Offer, Negotiation, Approval/Signing, et al.). The complexity of the study should be illustrated.

4) In figure 4 block-1, section 3.3.2 “Consent-based Compliance Verification on B2C Contracts” and section 3.3.3 “Consent-based Compliance Verification on B2B Contracts”, the author describes the consent-based compliance verification on B2C and B2B contracts. But in the figure, when (B2B!=”” and B2C!=”” and Consent!=””), it can get B2B data. Please checks if it is (B2B!=”” and B2C==”” and Consent!=””).

5) In this paper, the contract just can be created in tool. If the author has considered that how the existing contracts (e.g. paper contract and unstructured contract) are translated into RDF?

6)  In Fig. 6b, there are many nodes named by a long string of letters and numbers. The meaning and relationship of this KG representation is not explicit. The auther can explain what a node stands for, a contract, or a clause?

7) In section 5 “Evaluation”, the author presents the evaluation of the tool with a focus on performance of CCV compliance functionalities. However, the author just shows the response time of the tools. Please supplement the accuracy results of the CCV method. The authors should design an experiment to validate the correctness of the contract creation and audit.

Author Response

Dear Reviewer, 

We would like to express our gratitude for taking the time to review our article. We have updated the manuscript, taking your feedback into account. The details of the changes made are provided in the attached file. 

 Regards,  

Amar Tauqeer (On behalf of all the authors) 

Reviewer 2 Report

This paper presents a knowledge graph-based contract compliance verification (CCV) tool  to bind GDPR legal basis to data sharing contracts.

The paper is well structured and presented, the related work section shows several related studies and authors highlight the shortcoming of the described works  with respect to GDPR, the functionality of the CCV tool is demonstrated with use cases, and  a performance  evaluation is shown.  In general, it is an interesting research and it seems that it already has a real application in real scenarios.  However, there are some aspects that have to be improved in this  paper to reach a better quality to be published. 

The difference between the previous publication [23] of the authors should be explicitly clear in this paper. What was presented in [23] and what is new  in this research.

Section 3:

Line 287  What do you mean “The CVV tool” Do you mean CCV? Even though, what is the CCV tool? You have not described any tool until this part.

Section 4

Figure 6, to be more clear maybe it is better to put into circles the mane of classes instead of the IDs (or just a part of the IDs).

From line 546 to line 563 seems to not  be related with Reduction of non-required attributes of data subjects.

Actually, it is not clear how the Reduction of non-required attributes of data subjects is done. Is it just eliminating the variables that are not required? 

Why Section 4.2.2 is named Experiment Setup? That section describe the libraries used in the implementation of the CCV tool, isn’t it? 

Section 5

Line 652. I think Section 4.2.2 is not the experimental setup. An experimental setup  describes how the experiment will be done (only one client connected, several clients?),   the hardware and configuration used for the experiments,  summarizes the data and scenarios, summarizes the considerations taken, etc. 

Why execution time is the most important metric in this work? Why not the quality of the responses of the tool? Can be it measured? Or may be authors should justify why the total execution time is important by explaining when the CCV tool is executed? Every period? Each time a modification in the repository is done? Is it executed by demand? How many clients of the tool can be simultaneously connected?  It lacks these details in the escenarios. 

Why the total time is high (more than 13 minutes in some cases)? Are there many I/O accesses?   Maybe these times limit the scalability of the tool. Can these times be improved in somehow? May be a reflection and a future work proposal can be added in this regard.

667 to 671… References to figures do not correspond.

In Table 2, it could be good to highlight the biggest and lowest measures.

References:

For those references that are only links of projects or dataset or  repositories, maybe it is better to add their URL links  as footnotes instead of being references, such as [46] to [50].

Review carefully the list of references. For example, ref [7] has not the year of publication.

A deep proof reading is needed to improve the writing style. In the following there are some suggestions, but for sure there many other to take care:

Abstract

Please provide the meaning of the acronymous (GDPR, TOM, etc.)  the firs time you used it in the abstract, introduction and conclusions, which should be self-contained sections. 

e.g. —> e.g.,   (check the whole paper)

i.e. ==> i.e., (check the whole paper)

Introduction

Avoid orphan references. For example, the text in line 19 and the references in line 20, or in lines 48 and 49, 84 and 85; or references of sections (lines 116, 117,118,119). If you use latex, this is solved with the ~ (e.g., … EU citizens’ data~\cite{xxxx,yyyy} or Section~\ref{zzzz}).  Check the whole paper.

Line 48 organisations, departments, people and databases ==> organisations, departments, people, and databases (In English, there is a comma before the and of the last element in a list with more than two elements. Check the whole paper.

Line 51 Locating specific data and permissions for its sharing such as consent and contracts can be  both time-consuming ==> Locating specific data and permissions for its sharing, such as consent and contracts, can be both time-consuming 

Line 59 To do so organisations ==> To do so, organisations 

Line 67 Provide the meaning of TOMs

Line 123 They are elaborated into the following three subsections 2.1, 2.2, and 2.3 1respectively. ==> They are elaborated into the following subsections. 

Or

They are elaborated into the following Subsection 2.1, Subsection 2.2, and Subsection 2.3,  respectively. 

Line 144 Provide the meaning of KPI 

Line 164 Saas ==> SaaS

Line 240 we do not have discussions  ==> we or they?

Line 256 to 259:  What does mean the phrase? That you have taken ideas from all those works. The phrase seems to me incomplete. It should Strat with We consider earlier work focused on …. 

Or  Our bases were inspired by earlier work focused on …

Line 260 We followed the approach  [32] to construct ==> We followed the approach described in [32] to construct 

Line 263 and guidelines from [36]. ==> and the guidelines proposed in[36]. 

Line 272 In Figure 1 it is presented ==> In Figure 1, it is presented 

Line 273 The contract life-cycle divides into six stages  ==> The contract life-cycle consists of  six stages 

Line 278 which can also define in the second stage ==> which can also be defined in the second stage 

Line 294 (Figure. 2) ==> (Figure 2

Line 379 The listed rules shroud be introduced. For example: Following rules describe ….: 

In title of Figure 4 should be CCV instead of CVV

Line 501 CVV ==> CCV

Line 506 As an example, the part of the contract is used to perform CRUD operations such as contract creation, contract audit, contract read, etc.  ==> etc. is redundant when you use such as:

As an example, the part of the contract is used to perform CRUD operations, such as contract creation, contract audit, contract read. 

Line 522 Our tool implements the following TOMs, which are described in the following. 

==> Our tool implements the following TOMs: 

Line 533 (Art. 32 (1) (b))) ==> (Art. 32 (1) (b))

Line 540 (Art. 32 (1) (d)), (Art. 25 (1))). ==> (Art. 32 (1) (d), Art. 25 (1)).

Line 546.. The phrase is incomplete

Line 548 (Art. 32 (1) 548 (d)), (Art. 25 (1))) ==> (Art. 32 (1) (d), Art. 25 (1))

In title of Figure 15 should be CCV instead of CVV

Author Response

(The authors gave the same response as above.)

Reviewer 3 Report

The following should be noted and corrected accordingly:

1. How practicable is your proposed model in real-time? 

2. Is it cost-efficient?

3. Some diagrams and terms are not properly explained.

4. Grammar is not up to standard and requires extensive re-editing

5. Are the formulas and numbers here generic or generated by you?

Study and consider the following related papers to embellish your paper:

• DOI: 10.3390/electronics8111331

• https://doi.org/10.1002/ett.3997

• https://doi.org/10.1016/j.compeleceng.2021.106996.

Major revisions are required.

Author Response

(The authors gave the same response as above.)

Round 2

Reviewer 1 Report

The authors have carefully revised the manuscript according to the reviewer's suggestions.

Reviewer 2 Report

I think authors have improved properly the document. Only few more suggestion:

Line 59 :verification (CCV). In digital contracting or agreements, the 59 CCV is a process that ensures data processing according to GDPR by detecting contractual 60 conflicts or breaches. A contract breach or conflict is a failure without legal excuses, to 61 perform any promise that forms all or part of the contract [9];  ==> The red part can be show as a footnote

Line 320: and signed as-is; There may be substantial  ==> and signed as-is; there may be substantial